# Antidepressant Screening Demonstrated Non-Monotonic Responses to Amitriptyline, Amoxapine and Sertraline in Locomotor Activity Assay in Larval Zebrafish

**DOI:** 10.3390/cells10040738

**Published:** 2021-03-26

**Authors:** Michael Edbert Suryanto, Gilbert Audira, Boontida Uapipatanakul, Akhlaq Hussain, Ferry Saputra, Petrus Siregar, Kelvin H.-C. Chen, Chung-Der Hsiao

**Affiliations:** 1Department of Bioscience Technology, Chung Yuan Christian University, Chung-Li 320314, Taiwan; michael.edbert93@gmail.com (M.E.S.); gilbertaudira@yahoo.com (G.A.); anjumarman390@gmail.com (A.H.); ferrysaputratj@gmail.com (F.S.); siregar.petrus27@gmail.com (P.S.); 2Department of Chemistry, Chung Yuan Christian University, Chung-Li 320314, Taiwan; 3Department of Chemistry, Faculty of Science and Technology, Rajamangala University of Technology Thanyaburi, Thanyaburi 12110, Thailand; boontida_u@rmutt.ac.th; 4Department of Applied Chemistry, National Pingtung University, Pingtung 900391, Taiwan; 5Center for Nanotechnology, Chung Yuan Christian University, Chung-Li 320314, Taiwan

**Keywords:** zebrafish larvae, antidepressants, behavior, locomotion, phenomics, biphasic effect

## Abstract

Antidepressants are well-known drugs to treat depression and major depressive disorder for humans. However, the misuse and abuse of antidepressants keep increasing with several side effects reported. The aim of this study was to assess the potential adverse effects of 18 antidepressants by monitoring zebrafish larval locomotor activity performance based on the total distance traveled, burst movement count, and total rotation count at four dark-light intercalated phases. In general, zebrafish larvae displayed sedative effects after antidepressant exposure by showing a significant reduction in all of the locomotor activity-related endpoints. However, three antidepressants i.e., amitriptyline, amoxapine, and sertraline were able to trigger a significantly high locomotor activity in zebrafish larvae during the light cycle. These differences might be due to the pharmacologic differences among the antidepressants. In addition, since each antidepressant possesses a different dosage range from the other, overdoses of these antidepressants might also be the causes of these differences. Furthermore, based on these results, a further study was conducted to observe the effect of these three antidepressants in lower concentrations. From the results, biphasic effects in terms of zebrafish larval locomotor activity were demonstrated by these drugs. Even though further studies are still required to validate the mechanism, these findings indicate that these antidepressants might share a common mechanism responsible for their effects on zebrafish larval locomotor activity although there were some differences in potency of these effects.

## 1. Introduction

Depression is a common mental health condition associated with a state of sadness, hopelessness, and grief that may affect the thinking process, behavior, and feelings. The changes in mood, emotion, and behavior are due to chemical imbalances of neurotransmitters in the brain [1]. Depressed individuals are likely to have low levels of neurotransmitters [2]. Antidepressants can help to balance those chemicals by acting as regulators that inhibit the reuptake of one or several neurotransmitters (serotonin, dopamine, or noradrenaline) through selective receptors [3]. Thus, antidepressants increase the level of those specific neurotransmitters around the nerves in the brain and central nervous system (CNS) [1]. Antidepressant drugs have been used to treat depression and anxiety disorders for over six decades and it has been proved to be an effective solution to treat major depressive disorder, social withdrawal, low mood, anhedonia, and other psychiatric symptoms [4]. However, some literature reported the abuse of antidepressant drugs. Several cases were identified among individuals repeatedly using antidepressants without prescription or excess of prescribed dose [5,6,7,8,9]. Since antidepressants have psychoactive properties that create a pleasant or excited feeling, as well as elevated mood [10], this could be one of the reasons why people misuse antidepressants in order to get a temporary euphoria. Along with the psychoactive effects of antidepressants, they produce side effects which can also be fatal. Some studies discovered that the use of antidepressants led to seizure, self-poisoning, mortality, and an increase of suicidal attempts in humans [11,12,13]. Moreover, they also reported that it could produce sedation, agitation, insomnia, akathisia, apathy, and emotional blunting [14]. Several side effects from antidepressant treatment were also reported with common symptoms of insomnia, headaches, nausea, dry mouth, and sexual dysfunction. Yet, the full range of these adverse effects from antidepressants and their relation has not been elucidated.

Nowadays, zebrafish (*Danio rerio*) is a well-accepted animal model in various research fields including genetic studies, drug discovery, and behavioral analysis. In addition, it is frequently used for pharmacology safety investigations since it possesses several advantages over other animal models [15,16,17]. Firstly, it is extremely fertile and can produce progeny in large numbers, allowing scientists to collect and conduct experiments with high sample numbers [18,19]. Secondly, it is easily maintained in the laboratory with low associated costs and a rapid reproductive cycle [15,20]. Thirdly, it has a relatively small size, therefore, their locomotor activity can easily be observed and measured in high-throughput screening [21,22]. Furthermore, the neurochemistry of zebrafish also makes it a suitable model to test various drugs with neuroactive or psychotropic activity [22,23,24]. Lastly, this aquatic model organism possesses several neurotransmitters (serotonin, dopamine, noradrenaline, gamma-aminobutyric acid/GABA) that are important in regulating physiological and behavioral activities and possibly targeted by antidepressants [25]. Taken together, this animal model is a good representative for studying the adverse effect of antidepressants.

The present study aimed to generate systematical data for a deeper understanding of the adverse effects of various types of antidepressants based on animal behavior of zebrafish larvae. In this study, the zebrafish larval photomotor responses to an intense light–dark transition exposure were assessed since the previous study in amphipods, crustaceans, and fish found that antidepressant drugs induced the phototaxis behavior [26]. Behavioral approach has been used for drug discovery, pharmacological safety assessment, and neurotoxicity screening; such as anticonvulsant drugs, antipsychotics, anxiolytics, and antidepressants [27,28,29,30,31]. Locomotion is a complex behavior that involves an integrated response from brain, nervous system, and visual pathway [32]. These behavioral patterns and changes are predictive of observable adverse effects that are useful to provide high-throughput screening of neuroactive compounds [33,34]. Behavioral alterations have been used as the most sensitive early toxicity and stress indicators in the organism [35,36]. Thus, in this study, several antidepressant drugs that are well known and widely used were inspected by using a behavioral approach in larval zebrafish. The classification of these antidepressants, based on their working mechanisms and also the adverse side effects, are specified in Table 1. A 1 mg/L (ppm) concentration was applied in the current study and it was chosen based on literature review for the highest possible concentration of antidepressant that can be administered without causing severe damage or loss of life. A prior study showed that the lethal concentration (LC_50_)-96 h of selective serotonin reuptake inhibitor (SSRI) exposure in fish was within 3.29–6.34 mg/L [37]. In addition, another study revealed that the LC_50_ of amitriptyline exposure in zebrafish embryos is within mg/L concentration (2–100 mg/L). However, 1 mg/L concentration is still deemed to be safe as the mortality rate was below 25% [38]. Therefore, the antidepressant screening was performed at a concentration of 1 mg/L across the board. The hypothesis of this study is that the locomotor performance in zebrafish larvae will be altered by antidepressants even at a very low concentration. Furthermore, since some antidepressants showed distinct effects from the majority of tested antidepressants, it was intriguing to conduct a deeper study on those antidepressants. Thus, a further experiment to observe the dose-effect of these antidepressants was carried out by using several different lower concentrations, which were 0.001, 0.01, and 0.1 mg/L. The overview of experimental design and workflow is illustrated in Figure 1.

## 2. Materials and Methods

### 2.1. Antidepressant Drugs

Eighteen different antidepressants were used for this experiment (abbreviation and working mechanism for each compound are summarized in Table 1). Amitriptyline hydrochloride (AMY), amoxapine (AMO), atomoxetine hydrochloride (ATM), bupropion hydrochloride (BUP), doxepin hydrochloride (DOX) duloxetine hydrochloride (DLX), escitalopram oxalate (ESC), fluoxetine hydrochloride (FLX), imipramine hydrochloride (IMP), mianserin hydrochloride (MIA), milnacipran (MCP), moclobemide (MEM), and venlafaxine hydrochloride (VEN) were purchased from Shanghai Aladdin Bio-Chem Technology Co., Ltd., Shanghai, China; while mirtazapine (MRT) was purchased from Adamas Beta, Shanghai, China. Paroxetine (PAR) was purchased from Shanghai Bide Pharma Tech Co., Ltd., Shanghai, China. Selegiline hydrochloride (SEG) was purchased from Shanghai Yuanye Biotechnology Co., Ltd., Shanghai, China. Sertraline hydrochloride (SRT) was purchased from Shanghai Macklin Biomedical Co., Ltd., Shanghai, China. Trazodone hydrochloride (TRA) was purchased from Sigma-Aldrich Taiwan Merck Co., Ltd., Taipei City, Taiwan. All antidepressants were dissolved in organic solution 0.01% in acetone as 1000× stocks and stored at 4 °C.

### 2.2. Animal Housing and Ethics

The AB zebrafish stock was obtained from Taiwan Zebrafish Core Facility at Academia Sinica and kept in the laboratory. Sex-matured male and female wild type (WT) zebrafish aged 4–6 months were used for the breeding process. Afterward, the embryos were collected and placed into a 9 cm petri dish, filled with a mixture of E3 medium and methylene blue, which was used to prevent fungal infections, until a concentration of 0.1 mL/L was reached following our previous protocols [64,65]. Zebrafish embryos/larvae were incubated at 28 °C and the pH was maintained between 6.8 and 7.5 (Figure 1, top-left panel). The protocol for proper usage of zebrafish has been authorized by Institutional Animal Care and Use Committee (IACUC) at Chung Yuan Christian University (Approval No. 109001, issue date 20 January 2020). A supplemental light source was provided for 12 h of each light and darkness per day.

### 2.3. Antidepressants Exposure on Zebrafish Larvae

The antidepressants were diluted from the stock solution to working concentration (1 mg/L). Meanwhile, for further experiments, several lower concentrations (0.001, 0.01, and 0.1 mg/L) of selected antidepressants, which were AMY, AMO, and SRT were administered to the zebrafish larvae by immersion. Later, after 4 days post fertilization (dpf), zebrafish larvae were randomly selected for the treatment and control groups. For the control group, the sample of zebrafish was placed into a 9 cm petri dish filled with a 40 mL mixture of E3 medium and methylene blue. Meanwhile, for the treatment group, the zebrafish were placed into a 9 cm petri dish supplemented with a 40 mL mixture of E3 medium, methylene blue, and 1 mg/L antidepressant. Both the control and treatment groups were incubated at 28 °C with pH 6.8–7.5 for 24 h (Figure 1, middle-left panel). The experiments were performed in duplicate. The first antidepressant screening was conducted in three rounds with each round using one group of control (48 larvae) and the other six groups of antidepressant treatment (each group 48 larvae). Thus, a total of three control groups (144 larvae) and 18 antidepressant groups (864 larvae) were used in this screening. Further experiments of three selected antidepressants were performed with 96 larvae for each concentration of tested antidepressants with a total of 864 larvae, while 167 larvae were used for control groups.

### 2.4. Zebrafish Larvae Locomotion Test

After exposure to antidepressants for 24 h, the zebrafish larvae were individually transferred into a 48-well plate to be observed with ZebraBox (ViewPoint 3.22.3.85, Viewpoint Life Sciences, Inc., Civrieux, France) [66]. After 30 min of acclimation, zebrafish larvae locomotor activities were analyzed and measured with Viewpoint ZebraLab software integrated with the ZebraBox. Each locomotor activity test lasted for 80 min, which consisted of four light cycles and four dark cycles with 10 min duration for each cycle at room temperature. The locomotor activities were assessed by measuring three important locomotor endpoints; total distance traveled, burst movement, and rotation counts (Figure 1). For the total distance traveled endpoint, thresholds were set to differentiate their movement type during the test. These thresholds were as follows: large movement (>2 cm/s), small/normal movement (0.5–2 cm/s), and inactivity (<0.5 cm/s) [65]. Next, the burst movement count that indicated their cruising activity was set based on the pixel intensity changes in their body. The applied thresholds were 20 pixels or more for bursting and less than 5 pixels for freezing. Lastly, in the rotation count, clockwise and counterclockwise rotations per minute were counted throughout the test. The thresholds were adjusted based on the minimum diameter (5 mm) and 60° of back angle. Any rotation with a greater value than the minimum diameter and back angle was counted as one rotation count. The locomotion activity results are displayed for each integration period of 1 min. All the parameters for ZebraBox were set following previous publications [65,67].

### 2.5. PCA, Heatmap, and Clustering Analysis

To provide a better resolution for the adverse effects of antidepressants, the phenomic approach analysis was used to combine data collected from zebrafish locomotion alteration (Figure 1, bottom-right panel). Principal component analysis (PCA) and hierarchical clustering analysis were undertaken using ClustVis web tool (ClustVis version 2018, University of Tartu, Tartu, Estonia) [68,69]. Three different locomotor activity endpoints, namely (1) distance traveled, (2) burst movement count, and (3) rotation count in both light and dark cycles were used to generate a data matrix. Later, the average locomotor activity endpoint data were summarized and input into a spreadsheet using Microsoft Excel. After it was saved as a comma-delimited type file (.csv), it was uploaded to the ClustVis web tool. Next, to treat each variable for each row equally, a unit variance scaling was applied. To calculate principal components, singular value decomposition was utilized since there were no missing values in the dataset.

### 2.6. Statistical Analysis

Graphic results and statistical analyses were performed using GraphPad Prism (GraphPad Prism 8.0.2, GraphPad Software, Inc.: San Diego, CA, USA) [70], a scientific graphing and statistics software. Initially, the data distribution normality was analyzed by D’Agostino and Pearson, Saphiro–Wilk, and Kolmogorov–Smirnov test to determine the appropriate statistical analysis used in the following test. The locomotor endpoints in light and dark cycle statistical analysis tests, Mann–Whitney test, a pairwise nonparametric analysis, was conducted to compare the fish locomotor activity since the data were not normally distributed. Meanwhile, the total distance traveled, burst movement count, and rotation count per minute were analyzed by the two-way analysis of variance (ANOVA) with Geisser–Greenhouse correction to calculate the P-value from the dataset. Since each antidepressant group was compared individually to the control group, the multiple comparisons test as a post hoc test was not used in this study. The presented data are shown either with median with 95% CI or mean ± standard error of the mean (SEM) and significant differences were marked as * if *p* < 0.05, ** if *p* < 0.01, *** if *p* < 0.001, and **** if *p* < 0.0001. 

## 3. Results

### 3.1. Locomotor Activity Evaluation of Antidepressants Exposure in Zebrafish Larvae

For the locomotor activity test, the total distance traveled, burst movement, and rotation count of control and antidepressant-treatment groups were measured and compared. From the total distance traveled endpoint, it was found that the majority of the tested antidepressants altered zebrafish swimming activity patterns during light and dark cycles. Generally, the locomotor activity of normal larvae during the light cycle is lower than the dark cycle. This behavior creates a pattern of an elevated peak in the dark cycle followed by a recessed state in the light cycle. Here, this phenomenon was shown by the control group in Appendix A and it indicated that the zebrafish larvae from the control group exhibited a normal photomotor response. However, several antidepressant-treated groups did not show any remarkable difference between total distances traveled in both light and dark cycles, which in other words, diminished that constant pattern compared with control zebrafish larvae. In the dark cycle, most of the antidepressants significantly reduced the total distance activity (*p* < 0.0001), except MRT, which was not significantly different compared to the control group (*p* = 0.4438) (Figure 2B). Meanwhile, in the light cycle, not all of the treated groups displayed a significant reduction. These reductions were only displayed in MEM (*p* = 0.0004); DOX, SEG, MCP, VEN, BUP, MIA, and TRA treated groups (*p* < 0.0001) (Figure 2A). On the other hand, some treated groups including AMY, AMO, IMP, SRT (*p* < 0.0001); FLX (*p* = 0.0113); DLX *(p* = 0.0240); MRT (*p* = 0.0011) displayed an elevated locomotor activity and these increments were observed in several antidepressant-treated groups. In addition, no significant difference in locomotor activity was observed in the group treated with ESC (*p* = 0.8121), PAR (*p* = 0.4985), and ATM (*p* = 0.1652)-treated groups during the light cycle (Figure 2A). 

Further observation of the antidepressant effects in zebrafish larvae locomotor activity was conducted by analyzing their burst movement activity. Through this endpoint, it was confirmed that most of the antidepressants caused a sedative-like behavior. Overall, most of the antidepressants reduced the burst movement activity in the light cycle (Appendix A), which was supported by the significant reduction of burst movement count observed in 14 different treated groups, including DOX, MEM, SEG, FLX, PAR, DLX, MCP, VEN, ATM, BUP, MIA, MRT, TRA (*p* < 0.0001); IMP *(p* = 0.0002)-treated groups. Furthermore, it is interesting to find that AMY, AMO, and SRT significantly increased the burst movement activity in zebrafish larvae during the light cycle (*p* < 0.0001) (Figure 2C). On the other hand, ESC showed no significant difference (*p* = 0.6044) compared to the control group in this cycle. Similar to the light cycle, almost all burst movement activity from antidepressants-treated zebrafish was reduced during the dark cycle (Figure 2D). These significantly decreased burst movement activities were displayed by DOX, IMP, MEM, SEG, ESC, PAR, SRT, DLX, MCP, VEN, ATM, BUP, and MRT-treated groups (*p* < 0.0001) while no significant difference was shown in AMY (*p* = 0.4524) and FLX (*p* = 0.2383)-treated groups compared to control. On the contrary, a significant change in locomotor activity was exhibited by three different antidepressant groups; AMO, MIA, and TRA (*p* < 0.0001) (Figure 2D). Taken together, the results showed that most of the antidepressants tested could reduce the burst movement activity when antidepressants were given at 1 ppm concentration. 

Finally, the potential effect of antidepressants on movement orientation was investigated by calculating the rotation movement counted. Overall, average rotation movement in antidepressant treated groups was significantly reduced in both light and dark cycles (*p* < 0.0001) (Appendix A), except for the SRT and MRT-treated group, which showed a significantly high level of average rotation movement count during the light and dark cycles (*p* < 0.0001, *p* = 0.0002), respectively (Figure 2E,F). However, there was no significant difference observed in AMY (*p* = 0.7292), AMO (*p* = 0.5334), and ESC (*p* = 0.7079)-treated groups during the light cycle regarding their movement orientation. The details of the two-way ANOVA test results from these experiments can be found in Appendix A.

### 3.2. Analysis of Locomotion Alteration in Zebrafish after Exposure to Antidepressants by a Phenomic Approach

To evaluate the behavioral endpoint similarities triggered by different antidepressants, PCA and hierarchical clustering with information collected from the locomotor activity tests were performed. From the hierarchical clustering result, three major clusters were generated. BUP, MIA, TRA, FLX, MCP, and SEG were found to be grouped in a single major cluster while ATM, MRT, and ESC belonged to another major cluster with the control group in it. Lastly, another cluster, which contained most of the groups, consisted of DLX, MEM, SRT, IMP, PAR, DOX, VEN, AMY, and AMO (Figure 3A,B). The grouping of the first cluster was plausible since all of the antidepressants caused decrements in all of the zebrafish behavior endpoints in the light cycle, which was not shown in other groups. Meanwhile, based on the second cluster, it was concluded that ATM, MRT, and ESC caused minor changes in locomotor activity of zebrafish larvae since these three antidepressants displayed a similar heatmap pattern with the control group. Next, three minor clusters were observed in the last clusters. Mostly, all of the groups in these clusters affected the locomotor activity of zebrafish by reducing all of the behavioral endpoints during the dark cycle. Interestingly, there was one minor cluster that showed a unique pattern compared to other groups. This cluster consisted of AMY and AMO that displayed higher values in most of the behavioral endpoints in both cycles. In addition, based on the PCA result, several antidepressants, including AMY, SRT, AMO, MRT, and MIA, caused contrast behavioral effects compared to most of the antidepressants. Thus, a subsequent study was needed to further observe the effect of these antidepressants.

### 3.3. Locomotor Activity Evaluation of Amitriptyline, Amoxapine, and Sertraline in Three Different Concentrations

Based on the locomotor activity test results above, the three antidepressants, which were AMY, AMO, and SRT, caused distinct effects in zebrafish larvae locomotor activity compared to most of antidepressants tested. While other antidepressants caused hypoactivity in both cycles, these antidepressants led to hyperactivity in zebrafish larvae during the light cycle even though suppressed locomotion was still observed in the dark cycle. Thus, to do a further examination of their effects in different doses, zebrafish larvae were exposed with the other three lower concentrations and locomotor activity of zebrafish larvae was again conducted. Overall, both AMY and AMO at low concentrations, especially 1 and 10 ppb, caused hyperactivity in zebrafish larvae during both light and dark cycles. These phenomena were indicated by the significantly high levels of several behavior endpoints, especially total distance traveled (Figure 4A–L and Appendix A). Interestingly, fish exposed to the higher concentration showed a trend towards a decreased locomotor activity. This conclusion, however, does not mean that the 100 ppb-treated groups of both antidepressants already exhibited a significantly lower locomotor activity than the control group since some behavioral endpoints, which were burst movement and rotation counts, still showed significantly higher values in these treated groups compared to the control group. There is a possibility that these antidepressants affected the locomotor activity of zebrafish larvae in a dose-dependent manner and produced non-monotonic dose responses (NMDR). The NMDR was indicated by a clear biphasic pattern with inverted U-shaped curves displayed in the AMY and AMO-treated groups (Appendix A). Meanwhile, slightly different results were displayed by the SRT-treated groups. An indistinct hyperactivity-like behavior was observed in the lowest concentration group, which was supported by the high levels of burst movement and rotation counts even though their total distance traveled was at a similar level with the control group in both cycles (Figure 4M–R and Appendix A). However, the tendency to decrease locomotor activity by lowering the total distance traveled was already shown in the 10 ppb group and it was even more pronounced in the 100 ppb group. Thus, by combining these data with the results from Figure 2, which showed slightly higher locomotion in the 1 ppm-treated group, especially in the light cycle, SRT also produced a biphasic effect, however, as a U-shaped curve and in a dose-dependent manner (Appendix A). The details of the Kruskal–Wallis test in Figure 4 and the two-way ANOVA test in Appendix A results can be found in Appendix A, respectively.

## 4. Discussion

### 4.1. Acute Exposure of Antidepressants Altered Zebrafish Larvae Locomotor Activity

Zebrafish is well known as a sensitive bioindicator since it is able to respond to a large variety of chemicals [71]. Furthermore, this animal model has been used to assess psychoactive drug consequences in behavioral context [23,72,73]. In this study, the adverse effects of antidepressants were examined in the locomotor activity assay in zebrafish larvae. Generally, larval zebrafish are more active during the dark cycle than the light cycle. This phenomenon was indicated by an increased locomotor activity during light to dark cycle transition, followed by the decreased locomotion during the dark to light cycle transition [74,75]. The low locomotor activity pattern during the light cycle could be related to its transparent body and its natural predator-evasion. Meanwhile, the high activity during the dark cycle might be a result of zebrafish seeking night-time shelter [76]. Therefore, light and dark cycle conditions are important especially for studying antidepressant’s adverse effects because they might modify the behavioral patterns, anxiety responses, or even stress [74]. 

As expected, after exposure to the antidepressants, decrements in locomotion in zebrafish larvae were observed almost in all treated groups. This phenomenon is plausible since antidepressants work by targeting neurotransmitters in regions of the brain involved in neuroadaptive changes, which regulate the neurotransmission signaling cascades [77]. These neurotransmitters are involved in many functions of the CNS, responsible for mediating locomotor and behavioral activity. The imbalance of these neurotransmitters can lead to different psychiatric disorders [78]. As the main target, neurotransmitters level in the synaptic cleft can be increased by antidepressants since they can enhance neural transmission which results in an antidepressant effect (relaxation) [79]. Low locomotor activity in animal models is an example of a psychoactive effect of antidepressants. The changes in a behavioral reaction indicate that the bodies are impacted by these psychiatric drugs [80]. The altered behavior was also further demonstrated by the phototaxis responses. The previous study highlighted that antidepressant induces phototactic responses which are associated with serotonergic activity [81]. Acute administration of antidepressants has been reported to modulate light-induced responses in hamsters, rats, and mice [82,83,84]. The observed effects of antidepressants with decreased swimming activity on zebrafish larvae are also consistent with other recent studies [85,86,87]. Other behavioral changes were also displayed from burst and rotation movement that implied fear, predator avoidance, seizure, and movement orientation [88,89]. Burst count was considered as the rapid simultaneous movement, which helps evaluate the locomotion activity. The sudden increase in the acceleration of swimming movement is indicated as the startle response [74], which presented within the transition of light and dark cycles. Meanwhile, the rotation movement indicated a swimming orientation varied depending on various factors [64,89]. The variation in body rotation produces an escape kinematic response depending on the stimulus. This thigmotaxis behavior is crucial for studying anxiogenic and anxiolytic drugs [74]. The lower or decreased burst and rotation activity on animals serves as an anxiolytic-displayed behavior [90], which is demonstrated in this study. The hypoactivity is displayed as the predominant response in the larval stage fish due to the exposure of antidepressants that affected their spontaneous swimming behavior [91]. 

### 4.2. Several Antidepressants Induced a Hyperactive Response in Zebrafish Larval Locomotor Activity

However, while most antidepressants led to changes in animal behavior [92] and produced anxiolytic-effect that was indicated by lower or decreased locomotion activity [93,94,95], surprisingly, there were several antidepressants that increased zebrafish locomotor activity despite the same effective concentration and exposure time. From the results, AMY, AMO, IMP, FLX, SRT, and DLX were found to induce hyperactivity. However, more pronounced hyperactivity-like behavior in this cycle was displayed by AMY, AMO, and SRT since these groups also showed a significantly high level of burst movement count. These phenomena, however, were similar to a prior study in pentylenetetrazole, another neuroactive drug, which was reported to cause an instantaneous change in the zebrafish swimming activities in response to changes in lighting conditions [96]. In line with the current results, a prior study found that a high concentration of AMY (1 mg/L) resulted in a significant elevation of adrenocorticotropic hormone (ACTH). In humans, stimulation of ACTH release from the adrenal gland is accompanied by the hyperactivity of hypothalamo–pituitary–adrenocortical (HPA) axis that contributes to major depression development. Furthermore, the oxidative stress that might have occurred in exposed fish can also play a role in this alteration since a previous study found that 1 mg/L of AMY inhibited antioxidant capacity [38]. In addition, the alteration in the locomotor activity of zebrafish larvae might be related to the decrement of body length of zebrafish larvae after exposure to AMY that was mentioned in a prior study [38]. Interestingly, a similar effect was also found in larval fathead minnows (*Pimephales promelas*) after exposure to SRT, one of the SSRI [86]. Here, SRT was found to increase the locomotor activity together with an abnormal movement orientation when the light was present. Meanwhile, in dark conditions, it reduced locomotion. Our results are concordant with previously published reports, which showed that behavior alteration in fathead minnows with SRT exposure only appeared during light conditions. It suggests that SRT generates an anxiolytic effect during the light period [97]. In line with these findings, acute treatment with antidepressants also could increase locomotor activity [98] and reduce the immobility time in rats [99]. Some prior studies also demonstrated hyperactivity in mice with reducing immobility time and an increase in escape behavior after treatment with serotonergic and noradrenergic antidepressants [100,101]. Furthermore, a similar behavior was also observed in AMO, an N-demethylated dibenzoxazepine, -treated group. These differences could be linked to the pharmacological properties of AMO, which interferes with several neurotransmitters [102]. Moreover, when taken in overdose, AMO leads to several neurological effects, including coma, convulsions, irreversible brain damage, and sometimes death [103]. In addition, this drug seems to carry a risk of worsening motor function in human patients with Parkinson’s disease (PD) [104]. These alterations of locomotor activity are highly related to the antidepressant effect since antidepressants can exert their effects through brain-derived neurotrophic factor (BDNF), serotonin, dopamine, and noradrenaline alterations. These neurotransmitter alterations are the cause of the locomotion behavioral differences found in zebrafish larvae during antidepressants treatment [105]. Two possible manifestations may occur, one as hypolocomotion and the other as hyperlocomotion. Hypolocomotion is usually manifested as an anxiolytic effect of antidepressants, while hyperlocomotion indicates anxiogenic effect due to the toxicity of antidepressants [106].

### 4.3. Biphasic Pattern of Amitriptyline, Amoxapine, and Sertraline in Zebrafish Larvae Locomotor Activity

Interestingly, after the further investigation of these three antidepressants at lower concentrations, it was found that they displayed NMDR on locomotor activity in zebrafish larvae, that is, they can decrease and increase zebrafish larval locomotion resulting in a biphasic curve. In fact, even though they were infrequently observed, the biphasic effects of some antidepressants had been demonstrated in several prior studies. The biphasic effects of these antidepressant drugs have been clearly demonstrated on producing the spike activity in perfused guinea pig hippocampal slices with some drugs displaying an opposite effect to each other. These results indicated that some antidepressant drugs may exhibit both anticonvulsant and convulsant effects [107]. Furthermore, AMY has also been shown to have biphasic effects on aggressive behavior in isolated mice. In their study, it was found that a low dose of this antidepressant increased the mice’s aggressive behavior while, at a high dose, AMY significantly reduced their spontaneous motor activity [108]. In addition, several tricyclic antidepressants, including IMP, have also been reported to produce a biphasic effect on brain excitability in man and in a variety of laboratory models of epilepsy. At low doses, they showed antiseizure actions while at higher doses, convulsant effects were displayed by these drugs. The low-dose antiseizure effects of IMP may come from its mechanism that blocks the active reuptake of brain noradrenaline both in vitro and in vivo. Meanwhile, the anticonvulsant action that occurred at higher doses may be due to the direct depressant effects exerted by this antidepressant on both peripheral nerve and muscle that may be due to a reversible depression of potassium and sodium conductances [109]. Taken together, the observed biphasic effects of these tricyclic antidepressants might be related to the noradrenaline released from adrenergic nerve endings of the heart to the CNS and affects the locomotion of zebrafish larvae [110]. Next, SRT, a serotonin transporter inhibitor was also found to influence BDNF gene expression in a biphasic manner following repeated injections in the rat hippocampus [111]. Moreover, in another study in the brains and testes of mice, a biphasic effect of SRT was also demonstrated. In their study, they determined the time course of the inhibition of phopho-glycoprotein (P-gp), a plasma membrane ATP binding cassette-transporter of the multidrug resistance family, at the blood–testis barrier and blood–brain barrier by SRT. A P-gp substrate, was coadministered with SRT, and its accumulation in the brain and testes was increased after 5 min, but by 60 min, it was reduced compared to control while by 240 min, the accumulation was elevated again. It was proposed that the biphasic effect might be caused by SRT metabolite, desmethylsertraline, which can inhibit P-gp, and changes in P-gp-independent transport of digoxin [112]. In addition, the biphasic effect of SRT was also showed in the locomotor activity assay in rats. Over the first hour of a 3-h test session, an initial suppression of amphetamine-stimulated hyperactivity was observed after SRT treatments in three different doses. Meanwhile, a significant enhancement in the motor stimulatory effect was produced in the third hour of testing. This phenomenon was a possibility due to the induction of stereotypy, which would have resulted in reduced forward locomotion and could account for its enhancement on amphetamine-induced hyperactivity [113]. Taken together, all these antidepressants had biphasic effects on the locomotor activity of zebrafish larvae although some differences in potency of these effects were noted. Such a concentration–response relationship indicates that specific mechanisms are responsible for the various phases and that these mechanisms predominate at different tissue concentrations [114]. However, all of the proposed mechanisms in the current study are a matter of additional study. Importantly, one has to keep in mind that this biphasic action of these antidepressants may limit the drug’s clinical utility. Furthermore, our data indicate that these three antidepressants might share a common mechanism responsible for their effects on zebrafish larval locomotor activity that differs from the other tested antidepressants in the current study.

### 4.4. Possible Mechanisms of Antidepressant Effects and Limitations in This Study

As mentioned above, different effects between each antidepressant were also demonstrated in the current study. These phenomena likely occurred due to the pharmacological differences among the antidepressants and overdoses of these antidepressants since each of them has its own dosage range [115] and thus, it could have additional nonspecific alterations due to the high concentration level. Even though the antidepressant trial has been applied to treat patients with depression and epilepsy, however, some medications with rapid dose escalation and high doses are more likely to induce seizure activity, psychotic symptoms, tremor, cardiotoxicity, and cause much anxiety [8,90]. Other possibilities that some antidepressants may produce different effects are due to the pharmacologic differences, even among the same class of antidepressants; such as their chemical structure, half-life, and how efficiently they are metabolized [116]. A previous study demonstrated that behavioral responses to antidepressants were dependent on drug administration and cAMP response element-binding protein (CREB) function in the hippocampus. CREB phosphorylation is a downstream molecule targeted to a signal transduction pathway that is activated by antidepressants and it was affected differently based on the mechanisms of drug action [94]. Thus, this could be the reason for the slight differences of antidepressants in the way they affected neuroreceptors in the body since the psychoactive effect of antidepressants depends on their own unique and distinctive attributes. 

In addition, based on the heatmap clustering and PCA results, several antidepressants were grouped together with the control group in a single major cluster. This phenomenon indicated that these antidepressants, which were ESC, ATM, and MRT, caused less severe behavioral effects to zebrafish larvae than other antidepressants. In humans, the dosage ranges of these antidepressants are relatively lower than the other antidepressants, such as AMO, MEM, and BUP [117]. Thus, this result indicated that zebrafish larvae have different optimum dosage ranges of these antidepressants from humans. However, further studies in several different concentrations are required to confirm this result. Biochemical analysis is also required to investigate more specific mechanisms of antidepressant effects in this animal model’s neurotransmitters at either mRNA or protein levels. 

## 5. Conclusions

Taken together, this is the first study to address the adverse effects of multiple antidepressants based on locomotor activity and phototaxis behavior in zebrafish larvae by using three different endpoints: total distance traveled, burst, and rotation movement. This study proves that the behavioral approach is useful as an early screening tool to assess antidepressants. Overall, the results indicated that even though the majority of antidepressants produce anxiolytic-like effect in locomotion assay, there were several antidepressants that caused an anxiety-like behavior in zebrafish larvae, displayed by abnormal behavior with hyperactivity and altered phototaxis behavior. All of the behavior alterations caused by some antidepressants might be due to the pharmacologic differences among the antidepressants. In addition, the present study also found that several antidepressants, which were AMY, AMO, and SRT, exhibited biphasic effects, indicating that they can increase and decrease zebrafish larval locomotion in different concentrations. However, the results of this study do not rule out that other antidepressants might also have a biphasic effect since each antidepressant might have a specific range of effective concentration. The biphasic effect demonstrated by these three antidepressants might be related to the noradrenaline released from the adrenergic nerve and the induction of stereotypy by those drugs. Thus, further studies are required to investigate the specific mechanisms of antidepressant effects in different life stages of zebrafish.

## Figures and Tables

**Figure 1 cells-10-00738-f001:**
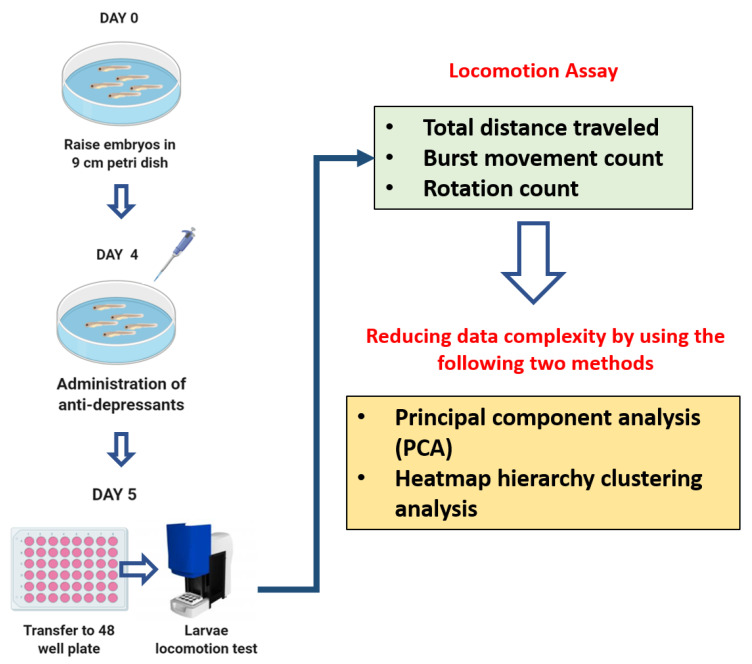
Schematic diagram of the experimental design to evaluate the potential adverse effects of 1 mg/L of 18 different antidepressants in zebrafish larvae. The experimental workflow for antidepressant exposure and locomotor activity measurement for larval zebrafish are illustrated in the upper panel. During locomotion assay, three major endpoints, including total distance traveled, burst count, and rotation count were measured and compared (top-right panel). Finally, two mathematic tools of principal component analysis and hierarchical clustering were used to reduce data complexity and perform similarity grouping (bottom-right panel).

**Figure 2 cells-10-00738-f002:**
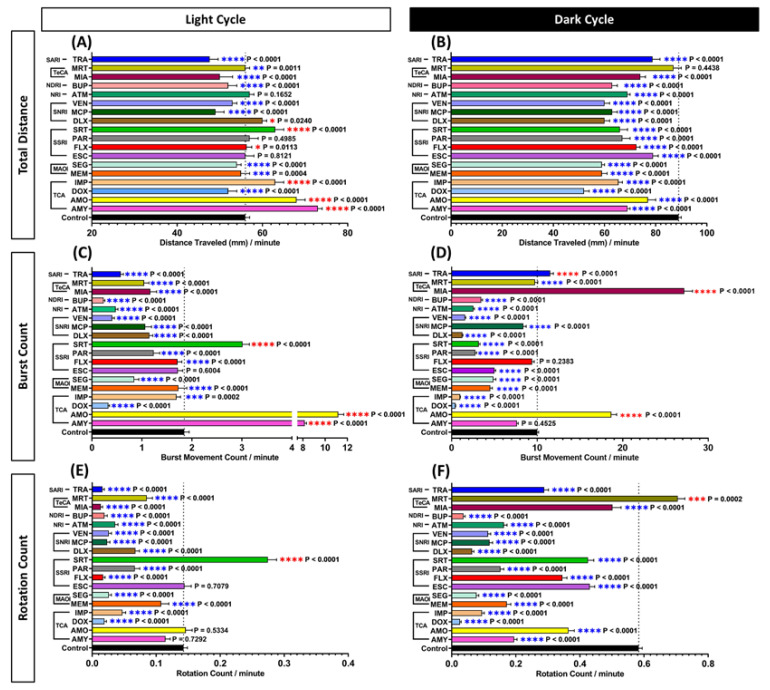
Comparison of (**A**,**B**) total distance traveled, (**C**,**D**) average burst movement, and (**E**,**F**) rotation count of zebrafish larvae in light and dark cycles, respectively. The statistical significance was tested by Mann–Whitney test. Every treated group was statistically compared to the control group individually. Data are presented as median with 95% CI for (**A**,**B**), and as mean with SEM for (**C**–**F**). Blue asterisk indicates a significant decrease in the value and red asterisk indicates a significant increase in the value in comparison to the control group (*n* = 144 for control, *n* = 48 for each tested antidepressant; * *p* < 0.05, ** *p* < 0.01, *** *p* < 0.001, **** *p* < 0.0001). The details of these two-way ANOVA test results can be found in Appendix A.

**Figure 3 cells-10-00738-f003:**
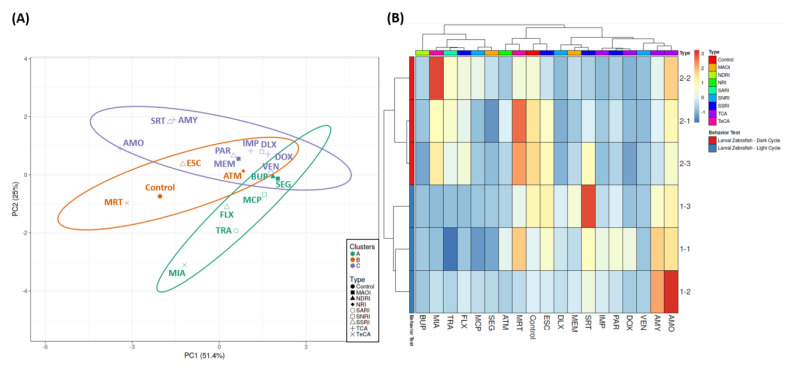
(**A**) Principal component analysis (PCA) and (**B**) hierarchical clustering analysis of locomotion behavior endpoints in zebrafish larvae after exposure to 1 mg/L of 18 different antidepressants. The untreated group is included as the control group. In (**A**), three major clusters from hierarchical clustering analysis results are marked with purple (1st cluster), orange (2nd cluster), and green (3rd cluster).

**Figure 4 cells-10-00738-f004:**
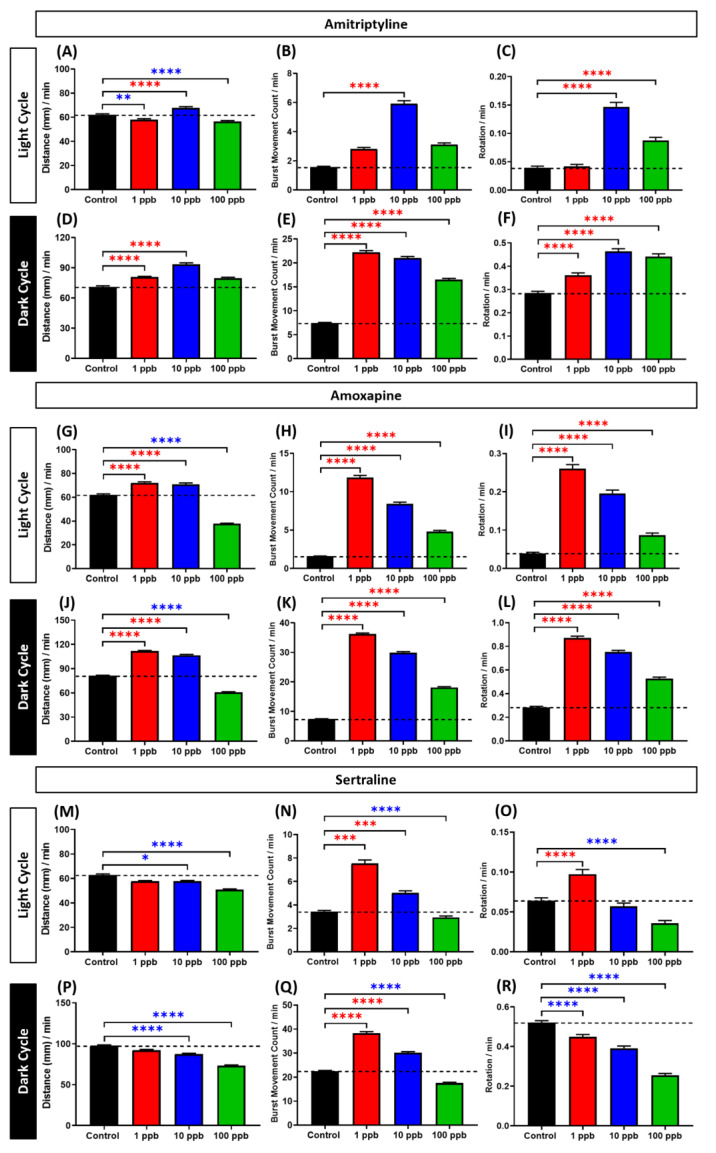
Comparison of (**A**,**D**,**G**,**J**,**M**,**P**) average total distance traveled, (**B**,**E**,**H**,**K**,**N**,**Q**) average burst movement, and (**C**,**F**,**I**,**L**,**O**,**R**) rotation count of zebrafish larvae after 1-day exposure of 0 (control), 1, 10, and 100 ppb of AMY, AMO, and SRT during both light and dark cycles. The statistical different significances between each treated group were statistically compared to the control group by Kruskal–Wallis test continued with Dunn’s multiple comparisons test. All of the data are presented as mean ± SEM. Blue asterisk indicates a significant decrease in the value and red asterisk indicates a significant increase in the value in comparison to the control group (*n* = 167 for control, *n* = 96 for each concentration of tested antidepressants group, except 10 ppb AMY group (*n* = 95), * *p* < 0.05, ** *p* < 0.01, *** *p* < 0.001, **** *p* < 0.0001). The details of these Kruskal–Wallis tests can be found in Appendix A.

**Table 1 cells-10-00738-t001:** List of antidepressants used in this study.

No.	Class	Mechanism	Name	Side Effects
1	TCA	Inhibit the reuptake of noradrenaline and serotonin [39]	Amitriptyline (AMY)	Dizziness, constipation, headache, and palpitations [40]
2			Amoxapine (AMO)	Seizures, severe metabolic acidosis, acute renal failure, and coma [41]
3			Doxepin (DOX)	Dry mouth, constipation, dizziness, tachycardia, grand mal seizure, tremor, and hyperthermia [42]
4			Imipramine (IMP)	Nausea, dizziness, and sedation [43]
5	MAOI	Inhibit monoamine oxidase enzymes (MAO-A/B) [44]	Moclobemide (MEM)	Insomnia, headache, nausea, agitation, diarrhea, and dizziness [45]
6			Selegiline (SEG)	Anorexia, musculoskeletal injuries, hallucinations, dyskinesia, cardiac arrhythmias, and orthostatic hypotension [46]
7	SSRI	Inhibit the reuptake of serotonin [47]	Escitalopram (ESC)	Ejaculation disorder, insomnia, diarrhea, dry mouth, somnolence, dizziness, hyperhidrosis, and fatigue [48]
8			Fluoxetine (FLX)	Sexual dysfunction, headache, and nausea [49]
9			Paroxetine (PAR)	Sexual dysfunction, weight gain, sleepiness, dry mouth, headache, and nausea [50]
10			Sertraline (SRT)	Agitation, insomnia, seizure, and sexual dysfunction [51]
11	SNRI	Inhibit reuptake of serotonin and noradrenaline [52]	Duloxetine (DLX)	Dry mouth, insomnia, fatigue, headache, nausea, dizziness, constipation, diarrhea, and hyperhidrosis [53]
12			Milnacipran (MCP)	Dry mouth, sweating, and constipation [54]
13			Venlafaxine (VEN)	Dry mouth, constipation, dizziness, diaphoresis, decreased libido, and induced acute dystonia [55]
14	NRI	Inhibit reuptake of noradrenaline [56]	Atomoxetine (ATM)	Hypertensive crisis, headache, abdominal pain, decreased appetite, vomiting, and nausea [57]
15	NDRI	Inhibit reuptake of noradrenaline and dopamine [58]	Bupropion (BUP)	Seizures, nonepileptic myoclonus, and confusion [59]
16	TeCA (NASSA)	Antagonizing α2-adrenergic and serotonin receptor [60]	Mianserin (MIA)	Periorbital edema [61]
17			Mirtazapine (MRT)	Induced nightmares and high incidence of somnolence [62]
18	SARI	Inhibit the reuptake of serotonin, noradrenaline, dopamine; antagonizing serotonin and α1-adrenergic receptor [60]	Trazodone (TRA)	Daytime sleepiness, headache, orthostatic hypotension, and drowsiness [63]

TCA: tricyclic antidepressant, MAOI: monoamine oxidase inhibitor, SSRI: selective serotonin reuptake inhibitor, SNRI: serotonin-noradrenaline reuptake inhibitor, NRI: selective noradrenaline reuptake inhibitor, NDRI: noradrenaline-dopamine reuptake inhibitor, TeCA (NASSA): tetracyclic antidepressant/noradrenergic and specific serotonergic antidepressant, SARI: erotonin antagonist and reuptake inhibitors.

## Data Availability

The original data presented in this study can be obtained from authors upon request.

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
