# Peer review of "Antidepressant Screening Demonstrated Non-Monotonic Responses to Amitriptyline, Amoxapine and Sertraline in Locomotor Activity Assay in Larval Zebrafish"

_cells, 2021, doi:10.3390/cells10040738_

Round 1
Reviewer 1 Report
This work uses zebrafish larvae to examine potential adverse effects of 18 antidepressants. Based on different locomotion tests, including total distance traveled, burst movement, and rotation counts, authors concluded that most of these antidepressants caused reduced locomotor activity, but three of them displayed biphasic effects. Adverse effects of these antidepressants have been tested and reported in other systems. Compared to previous works, results reported here are still too preliminary for understanding behavioral changes after these pharmacological treatments. Specific points are as follows:
1. It is not obvious to observe biphasic effects for amitriptyline, amoxapine, and sertraline during light and dark cycles. This conclusion is not convincingly supported by experimental data. Analyses using different locomotion tests did not show consistent biphasic responses of zebrafish larvae to low and high concentrations of antidepressants. For example, at least burst movement and rotation seem to be increased during light and dark cycles at all concentrations of amitriptyline and amoxapine, although differences in intensity could be observed. Total distance and rotation are clearly decreased during dark cycle following treatment by sertraline at different concentrations. Therefore, results presented in figure 4 are rather confusing, and not sufficiently explained and discussed in the text. It is also quite difficult to understand how biphasic effects are determined.
2. A related issue concerns the results in statistical analyses presented in figure 4. In several conditions, it is particularly difficult to know whether the values represent an increase or a decrease. These include but are not limited to 100ppb in figure 4A and 10ppb in figure 4M. No colored asterisks were used to differentiate increase and decrease in the values, as they did in figure 2. Although the calculated values were considered as significant, with only subtle differences between control and treated groups in many conditions, it is hard to evaluate the biological significance of these data.
3. Due to the weakness mentioned above, the title of this manuscrit is misleading.
4. Zebrafish embryos are generally kept in E2 medium, is ddH2O appropriate for culturing early embryos?
5. Figure 3 legend, “In Fig. 5A” should be “In Fig. 3A”.
Author Response
Comments and Suggestions for Authors
This work uses zebrafish larvae to examine potential adverse effects of 18 antidepressants. Based on different locomotion tests, including total distance traveled, burst movement, and rotation counts, authors concluded that most of these antidepressants caused reduced locomotor activity, but three of them displayed biphasic effects. Adverse effects of these antidepressants have been tested and reported in other systems. Compared to previous works, results reported here are still too preliminary for understanding behavioral changes after these pharmacological treatments. Specific points are as follows:
- It is not obvious to observe biphasic effects for amitriptyline, amoxapine, and sertraline during light and dark cycles. This conclusion is not convincingly supported by experimental data. Analyses using different locomotion tests did not show consistent biphasic responses of zebrafish larvae to low and high concentrations of antidepressants. For example, at least burst movement and rotation seem to be increased during light and dark cycles at all concentrations of amitriptyline and amoxapine, although differences in intensity could be observed. Total distance and rotation are clearly decreased during dark cycle following treatment by sertraline at different concentrations. Therefore, results presented in figure 4 are rather confusing, and not sufficiently explained and discussed in the text. It is also quite difficult to understand how biphasic effects are determined.
The authors appreciate the reviewer’s suggestion. To help the readers to understand the biphasic effect shown in the results, another figure of the locomotion activity percentage of the treated group compared to the untreated group was created and it is shown in this revised version. Based on the figure S3, the amitriptyline treated group displayed hyperlocomotion on mostly locomotor endpoints at 10 ppb concentration, except burst movement during a dark cycle which revealed hyperlocomotion at 1 ppb concentration; then it followed by decreased locomotion at 100 ppb concentration. The biphasic patterns are quite consistent with an abrupt increase in locomotor activity followed by a transient decrease before reaching the plateau effect. A consistent biphasic pattern was also clearly shown in the amoxapine treated group with a sudden increase of activity at 1 ppb concentration on all locomotor endpoints during light and dark cycles. Nevertheless, the authors agreed with the reviewer regarding the sertraline treated group that displayed a slightly different pattern compared to the TCA antidepressants. However, the sertraline-treated group still displayed a biphasic pattern with an inverted U-shaped curve, which was indicated by a slightly increased locomotion activity after treated with 1 ppm concentration. Furthermore, except for burst movement during the dark cycle, a sudden increase in burst activity was displayed in the lowest concentration and followed by decrements of locomotion activity along with the increase of concentration. The biphasic effects in this study were determined by the dynamic changes of locomotion activity with at low concentration can induce hyperactivity, while at high concentration can induce hypoactivity or vice versa. This term is also known as non-monotonic dose responses (NMDR), which is different from monotonic dose-response that only displays a linear curve. A complex NMDR also tends to produce a biphasic or multiphasic curve with a combination of U-shaped and/or inverted U-shaped curve toward varying concentrations. In addition, the biphasic effect observed in the current study was also in line with a previous study that stated the SSRIs therapy showed a complex and multiphasic process in reducing the cortisol level. Furthermore, another study also revealed that short-term exposure of TCAs displays non-monotonic, hormetic dose responses on snail behavior. The additional information regarding this matter had added to the manuscript.
Guler, Yasmin, and Alex T. Ford. "Anti-depressants make amphipods see the light." Aquatic Toxicology 99.3 (2010): 397-404.
Cvrčková, Fatima, Jiří Luštinec, and Viktor Žárský. "Complex, non-monotonic dose-response curves with multiple maxima: Do we (ever) sample densely enough?." Plant signaling & behavior 10.9 (2015): e1062198.
Dziurkowska, Ewelina, Marek Wesolowski, and Maciej Dziurkowski. "Salivary cortisol in women with major depressive disorder under selective serotonin reuptake inhibitors therapy." Archives of women's mental health 16.2 (2013): 139-147.
Fong, Peter P., et al. "Short-term exposure to tricyclic antidepressants delays righting time in marine and freshwater snails with evidence for low-dose stimulation of righting speed by imipramine." Environmental Science and Pollution Research 26.8 (2019): 7840-7846.
- A related issue concerns the results in statistical analyses presented in figure 4. In several conditions, it is particularly difficult to know whether the values represent an increase or a decrease. These include but are not limited to 100ppb in figure 4A and 10ppb in figure 4M. No colored asterisks were used to differentiate increase and decrease in the values, as they did in figure 2. Although the calculated values were considered as significant, with only subtle differences between control and treated groups in many conditions, it is hard to evaluate the biological significance of these data.
The authors fully understood the reviewer’s point. Initially, the data were expressed as the median with interquartile range since it is a proper presentation used if data are not normally distributed. However, as the reviewer mentioned, the authors fully consider that this issue might be confusing to the readers. Therefore, the data presented in Figure 4 has been updated with colored asterisks to differentiate the significant decrease or increase in the values similar in figure 2. In addition, the expressed data in figure 4 has also been changed to be presented as mean ± SEM to display a clearer representation of the results.
- Due to the weakness mentioned above, the title of this manuscript is misleading.
Thank you for pointing out this matter. As explained above and as the reviewer’s suggestion, the title of this manuscript was changed to avoid misleading.
- Zebrafish embryos are generally kept in E2 medium, is ddH2O appropriate for culturing early embryos?
The authors thank the reviewer for pointing out this matter. There was a mistake regarding the medium used for keeping the zebrafish embryos. Actually, the zebrafish embryos were maintained in an E3 medium with 0.1 mL/L of methylene blue to prevent fungal infections. Therefore, the authors had revised the methods, specifically in the animal housing and ethics section (line 136-138), according to these changes.
- Figure 3 legend, “In Fig. 5A” should be “In Fig. 3A”.
Thank you very much for the correction. A correction has been made regarding this issue.
Reviewer 2 Report
General
The authors describe the potential adverse effects of 18 antidepressants by monitoring zebrafish larval locomotor activity performance based on the total distance travelled, burst movement count, and total rotation count at four dark-light intercalated phases. Among these antidepressants, some of them amitriptyline, amoxapine, and sertraline were used in three different concentrations showing a biphasic effect. The paper is in general well written, however it needs some improvement.
Methods
Indicate the total number of larvae used in the present study.
Statistical section
“Mann Whitney test, a pairwise non parametric analysis, was conducted to compare the fish locomotor activity since the data were not normally distributed”. How was confirmed that the distribution was not normal?
Results
1)Why the most antidepressants were used at only one concentration? Is it possible that using increasing concentrations they could have shown a biphasic effect? Please provide an explanation.
2)Figure 4:panel A, D,G,J, M and P: the mean and mean standard error of some histograms are completely superimposable and it is very difficult to accept some statistical difference. Perhaps the authors used standard deviation and not mean standard error. Please clarify.
Discussion
It is not clear if the aim of the study was to verify the adverse effects of multiple antidepressants based on locomotor activity in zebrafish larvae or to demonstrate that it is possible to screen antidepressants on the basis of the locomotor activity alteration. In this last situation locomotor activity is not an index of toxicity only for antidepressants but also for many other categories of drugs acting on CNS. Furthermore, the antidepressant effect is not evaluated in this study and we cannot exclude that the alteration of locomotor activity is unrelated to the antidepressant effect. The authors are invited to provide a comment.
The discussion is too long and sometime speculative.
Author Response
Comments and Suggestions for Authors
General
The authors describe the potential adverse effects of 18 antidepressants by monitoring zebrafish larval locomotor activity performance based on the total distance travelled, burst movement count, and total rotation count at four dark-light intercalated phases. Among these antidepressants, some of them amitriptyline, amoxapine, and sertraline were used in three different concentrations showing a biphasic effect. The paper is in general well written however it needs some improvement.
Methods
Indicate the total number of larvae used in the present study.
The authors appreciate the reviewer’s suggestion. The first antidepressant screening was carried out in three rounds with one untreated group and six antidepressant treated groups (each group consisted of 48 larvae) for each round. Thus, a total of three untreated groups (144 larvae) and 18 antidepressant treated groups (864 larvae) were used in this screening. Later, further experiments of three selected antidepressants were performed with 96 larvae for each concentration of tested antidepressants group with a total of 864 larvae and 167 larvae for untreated groups. To sum up, for the whole experiment in this study; approximately 2039 zebrafish larvae were used. The authors had updated the manuscript, specifically on the antidepressant exposure on zebrafish larvae section, regarding this matter (line 173-178).
Statistical section
“Mann Whitney test, a pairwise non parametric analysis, was conducted to compare the fish locomotor activity since the data were not normally distributed”. How was confirmed that the distribution was not normal?
Thank you for your question. The normality distribution of the locomotor activity data was analyzed by using Normality and Lognormality Tests (Gaussian distribution) in GraphPad Prism prior to the further statistical analyses. In these analyses, several tests for testing their distribution normality, such as D’Agostino & Pearson, Shapiro-Wilk, and Kolmogorov-Smirnov test were conducted. Later, it was found that none of the data passed the normality test, thus the nonparametric tests, including Mann-Whitney test, were applied in this study. The authors had updated the Materials and methods section, specifically in the statistical analysis section (line 218-220), regarding this matter.
Results
1)Why the most antidepressants were used at only one concentration? Is it possible that using increasing concentrations they could have shown a biphasic effect? Please provide an explanation.
The authors appreciate the reviewer’s question. In the current study, most antidepressants were tested at only one concentration because the aim of this study is to conduct a screening of antidepressants that could give different effects among the others regarding zebrafish larvae locomotion. Therefore, based on the prior studies mentioned in the introduction section, only one particular concentration, which was 1 mg/L, was chosen as the concentration used in this screening. However, since only some of the antidepressants in this particular concentration that caused unusual behavior in the first screening were selected to be further evaluated, the authors fully aware that there is a possibility of other antidepressants to also have a biphasic effect that has not been observed yet since each antidepressant has a different effective concentration. Therefore, these matters become the limitations of the present study and future studies are required to be conducted. Thus, to enhance the manuscript’s quality, this crucial information had added to the conclusion section.
2)Figure 4: panel A, D,G,J, M and P: the mean and mean standard error of some histograms are completely superimposable and it is very difficult to accept some statistical difference. Perhaps the authors used standard deviation and not mean standard error. Please clarify.
Thank you for the suggestion. Actually, there was a mistake regarding the information of data expression used in Figure 4. Therefore, the authors had changed the data visualization into the mean and standard error of mean (SEM) so the error bars are not completely superimposable. Furthermore, in order to easily observe the statistical difference between untreated and treated groups, Figure 4’s statistical difference indicators were also colored, which also applied to Figure 2, to differentiate the increment and decrement in the values.
Discussion
It is not clear if the aim of the study was to verify the adverse effects of multiple antidepressants based on locomotor activity in zebrafish larvae or to demonstrate that it is possible to screen antidepressants on the basis of the locomotor activity alteration. In this last situation locomotor activity is not an index of toxicity only for antidepressants but also for many other categories of drugs acting on CNS. Furthermore, the antidepressant effect is not evaluated in this study and we cannot exclude that the alteration of locomotor activity is unrelated to the antidepressant effect. The authors are invited to provide a comment.
The authors understand the reviewer’s point of view. Actually, the aim of this study is to verify the adverse effects of multiple antidepressants based on locomotor activity in zebrafish larvae as the reviewer firstly mentioned. Thus, to achieve this objective, the current study used the developing zebrafish as a model system to examine the effect of the antidepressants on locomotor activity, which is based on three locomotor endpoints, namely total distance traveled, burst, and rotation movement counts; adapted with the photomotor response assay. However, regarding the last situation mentioned by the reviewer, here, the authors just wanted to give some explanations that might add the possibility to demonstrate an antidepressants screening based on zebrafish larvae locomotion activity. Oliveira et al. evaluated the exposure of nortriptyline, a tricyclic antidepressant, on zebrafish larvae locomotor behavior, which capable of impairing zebrafish swimming behavior (total swimming distance and time). In addition, another study by Richendrfer & Creton also used a behaviors approach (swim speed and percent of larvae at rest) in zebrafish larvae to analyze the antidepressant effect. Compared to these previous researches, the present study improved the locomotion endpoints with burst and rotation movement in both light and dark cycles; which are useful endpoints that indicate startle response, thigmotaxis, and sleep/wake behavior representative to help study the antidepressant effects. Furthermore, the alteration of locomotor activity was found to be highly related to the antidepressant effect, since antidepressant can exert their effects through brain-derived neurotrophic factor (BDNF), serotonin, dopamine, and noradrenaline alterations. These neurotransmitter alterations are the cause of the locomotion behavioral differences found in zebrafish larvae during antidepressant treatment. Two possible manifestations could occur, one as hypolocomotion and the other imply hyperlocomotion. The hypolocomotion usually manifests as the effects of antidepressants on anxiety-like behavior, while the hyperlocomotion is indicated as anxiogenic due to the antidepressant toxicity. Taken together, the last situation is not the current’s study most appropriate aim, therefore, the aim of this study was rephrased to be clearer and more suitable with the first situation mentioned by the reviewer.
Basnet, Ram Manohar, et al. "Zebrafish larvae as a behavioral model in neuropharmacology." Biomedicines 7.1 (2019): 23.
Herculano, Anderson Manoel, and Caio Maximino. "Serotonergic modulation of zebrafish behavior: towards a paradox." Progress in Neuro-Psychopharmacology and Biological Psychiatry 55 (2014): 50-66.
Oliveira, Ana C., et al. "Exposure to tricyclic antidepressant nortriptyline affects early-life stages of zebrafish (Danio rerio)." Ecotoxicology and Environmental Safety 210 (2021): 111868.
Richendrfer, Holly, and Robbert Creton. "Cluster analysis profiling of behaviors in zebrafish larvae treated with antidepressants and pesticides." Neurotoxicology and teratology 69 (2018): 54-62.
The discussion is too long and sometime speculative.
Thank you for the suggestion. The authors had tried their best to make the discussion part concise enough without neglecting the necessary information of this research. However, if the reviewer believes that there are still some unnecessary part in Discussion, the authors are opened to any suggestion. Furthermore, regarding the speculative matters, the authors consider that it is plausible to have some speculations since this is an initial study. However, these speculations, which are based on the current findings, are the foundations for future researches and might help the upcoming experiment’s study design.
Round 2
Reviewer 1 Report
This revised manuscript is improved after modifications in Figure 4 and in the main text. There are still a number of issues that need to be addressed, including but not limited to the following:
1. Lines 373-374, “The details of the Kruskal-Wallis test in Figure 4 and the two-way ANOVA test in Figure S2 results can be found in Table S2 & S3, respectively”. In Figure 4G, however, the statistical significance (P values) does not seem to correspond to the data shown in Table S3. The authors should clarify this issue and modify the text accordingly. Also, the authors need to clearly indicate the corresponding figures in the caption to different tables.
2. Line 42, “by act” should be “by acting”.
3. Line 61, “led to cause” should be either “led to” or “caused”.
4. Line 8, “Behavioral approach has been used to assessed drug discovery” may be “Behavioral approach has been used to assess drug discovery”.
5. Lines 159-160, “mixture of ddH2O and methylene blue” has not been changed.
6. Line 352, “combined with the results from 1 Figure 2”, there is a problem with this sentence.
7. Lines 393-395, lines 423-424, line 481, lines 514-516, these sentences should be checked for possible grammatical errors.
8. “fathead minnows” should written in a consistent manner.
Author Response
This revised manuscript is improved after modifications in Figure 4 and in the main text. There are still a number of issues that need to be addressed, including but not limited to the following:
- Lines 373-374, “The details of the Kruskal-Wallis test in Figure 4 and the two-way ANOVA test in Figure S2 results can be found in Table S2 & S3, respectively”. In Figure 4G, however, the statistical significance (P values) does not seem to correspond to the data shown in Table S3. The authors should clarify this issue and modify the text accordingly. Also, the authors need to clearly indicate the corresponding figures in the caption to different tables.
The authors appreciate the reviewer’s suggestion. Actually, there was a mistake regarding the statistical significance (P value) in Table S3. The authors had mistakenly put the P value of total distance AMY 100 ppb (Dark Cycle) in the column AMO 100 ppb (Light Cycle). Therefore, the authors had updated the P value with the correct one. All captions have been updated to clearly indicate the corresponding figures or tables. Thank you very much for your detailed review of our manuscript.
- Line 42, “by act” should be “by acting”.
Thank you very much for the correction. A correction has been made according to reviewer’s suggestion.
- Line 61, “led to cause” should be either “led to” or “caused”.
Thank you so much for your suggestion. The change has been made in the revised manuscript accordingly.
- Line 87, “Behavioral approach has been used to assessed drug discovery” may be “Behavioral approach has been used to assess drug discovery”.
The authors appreciate the reviewer’s suggestion. The correction has been made in the revised manuscript accordingly.
- Lines 159-160, “mixture of ddH2O and methylene blue” has not been changed.
Thank you very much for the correction. A correction has been made regarding this issue.
- Line 352, “combined with the results from 1 Figure 2”, there is a problem with this sentence.
Thank you very much for the correction. The authors had mistakenly put number 1 in the sentence. The authors also have restructured the sentence for better understanding.
- Lines 393-395, lines 423-424, line 481, lines 514-516, these sentences should be checked for possible grammatical errors.
The authors appreciate the reviewer’s suggestion. Therefore, the authors have restructured the sentence and grammatical check as the following:
On lines 393-395: This phenomenon is indicated by an increased locomotor activity during light to dark cycle transition, followed by the decreased locomotion during the dark to light cycle transition.
On lines 423-424: Meanwhile, the rotation movement indicated a swimming orientation varied depending on various factors. The variation in body rotation produces an escape kinematic response depending on the stimulus.
On line 481: Two possible manifestations may occur, one as hypolocomotion and the other as hyperlocomotion. Hypolocomotion is usually manifested as an anxiolytic effect of antidepressants, while hyperlocomotion indicates anxiogenic effect due to the toxicity of antidepressants.
On lines 514-516: These results indicated that some antidepressant drugs may exhibit both anticonvulsant and convulsant effects.
- “fathead minnows” should written in a consistent manner.
Thank you very much for the correction. The “fathead minnows” had been written in a consistent manner without using the capitalized letter.